# Maxwell Field of a Charge in Hyperbolic Motion

**Ramón Serrano Montesinos** [1,*] and **Juan Antonio Morales-Lladosa** [1,2]

1   Departament d'Atronomia i Astrofísica, Universitat de València, 46100 Burjassot, Spain
2   Observatori Astronòmic, Universitat de València, 46980 Paterna, Spain; antonio.morales@uv.es
*   Correspondence: rasemon@alumni.uv.es

**Abstract:** We conduct a detailed study of the electromagnetic field produced by a massive point particle undergoing hyperbolic (uniformly accelerated) motion in Minkowski space-time. Starting from the Liénard–Wiechert solution and using a covariant notation, we obtain and analyse the main quantities that describe this field. We identify the space-time region to which this solution is restricted and write a solution valid in the whole of space-time. Finally, we verify that this solution satisfies Maxwell's equations in the sense of distributions.

**Keywords:** Liénard–Wiechert potential; relativistic hyperbolic motion; Maxwell equations; tensor distributions

## 1. Notation

The physics described in this paper takes place in Minkowski space-time, a four-dimensional affine space endowed with a Lorentzian metric $\eta$ ($det\,\eta < 0$) with null curvature tensor (flat). The chosen signature is $(- + + +)$.

Tensors, whether expressed in contravariant or covariant components, including 4-vectors and 1-forms, will be denoted by Latin letters (such as $F$, $T$ or $e, h, s$). To avoid the risk of confusing a vector with a coordinate, position vectors (x, x′, y), the charge 4-velocity (u′) and the 4-acceleration (a′) will be shown in Roman font (x is the position 4-vector and $\{t, x, y, z\}$ the Cartesian coordinates). $u = t - x$ and $v = t + x$ will denote inertial null coordinates. Vectors in three dimensions, referred to an inertial observer, will be distinguished with an arrow (such as $\vec{E}$ and $\vec{H}$).

Tensor indices will be indicated by Greek letters. However, $\eta$ will always be reserved for the metric tensor and $\rho$ and $\chi$ for the cylindrical coordinates on the $(y, z)$ plane.

We denote by $\mathrm{u}_\flat = -dt$ the one-form metrically equivalent to the inertial observer $\mathrm{u}^\sharp = \partial_t$. From now on, the symbols $\flat$ and $\sharp$ will be suppressed (and we will write $\mathrm{u} = -dt$ and $\mathrm{u} = \partial_t$) because, in general, the covector or vector character is clear from the context or does not need to be specified. More generally, a sole and same letter will denote a vector and its metrically equivalent covector without distinction.

A point above a coordinate will indicate the derivative with respect to the proper time $\tau$ (as in $\dot{t}'$).

The partial derivative of a coordinate $u$ will be denoted by $\partial_u$ ($= \frac{\partial}{\partial u}$), the exterior differential of a 1-form $A$ or a 2-form $F$ will be denoted by $dA$ or $dF$, the symmetrized tensor product by $\widetilde{\otimes}$ ($\mathrm{x}\widetilde{\otimes}\mathrm{y} = \mathrm{x} \otimes \mathrm{y} + \mathrm{y} \otimes \mathrm{x}$) and the exterior product by $\wedge$ ($\mathrm{x} \wedge \mathrm{y} = \mathrm{x} \otimes \mathrm{y} - \mathrm{y} \otimes \mathrm{x}$). If $\alpha$ is a $p$-form and $\beta$ a $q$-form, an interesting property is $d(\alpha \wedge \beta) = d\alpha \wedge \beta + (-1)^p \alpha \wedge d\beta$.

The dual (Hodge) operator is denoted by $*$. Acting on a 2-form $F$, $[*F]_{\alpha\beta} = \frac{1}{2}\omega_{\alpha\beta\gamma\delta}F^{\gamma\delta}$, with $\omega_{\alpha\beta\gamma\delta} = -\sqrt{|det\,\eta_{\mu\nu}|}\epsilon_{\alpha\beta\gamma\delta}$ the metric volume element, $det\,\eta_{\mu\nu}$ the determinant of the metric tensor $\eta$ in the coordinate system considered $\{x^\alpha\}_{\alpha=0}^3$ and $\epsilon_{\alpha\beta\gamma\delta}$ the Levi-Civita symbol with the convention $\epsilon_{0123} = 1$.

To denote the contraction of a vector u with a 2-tensor tensor $F$ we will write $i(\mathrm{u})F$ if it takes place from the left ($[i(\mathrm{u})F]_\beta = \mathrm{u}^\alpha F_{\alpha\beta}$) and $F(\mathrm{u})$, from the right ($[F(\mathrm{u})]_\alpha = \mathrm{u}^\beta F_{\alpha\beta}$). We will denote the left and right contraction by $T(\mathrm{u}, \mathrm{u}')$ ($= \mathrm{u}_\alpha \mathrm{u}'_\beta T^{\alpha\beta}$).

The system of units is the *natural* one, where the speed of light in vacuum $c = 1$ and where Maxwell's Equations (our starting point) are written $dF = 0$ and $\nabla \cdot F = -J$, with $F$ the 2-form of the electromagnetic field and $J$ the 4-current.

## 2. Introduction

The problem of the electromagnetic field produced by a massive point charge in arbitrary motion, not subjected to an external electromagnetic field, was solved within a few years of the publication of Maxwell's theory by A. Liénard [1] and, independently, by E. Wiechert [2]. M. Born [3] was the first to particularise the problem to uniformly accelerated motion by calculating the field produced by a point electron in hyperbolic motion. Since then and up to the present day, several authors have dedicated their time and effort to this problem, focusing on its various complications. One of them concerns the validity of the field obtained from the Liénard–Wiechert potential. G. A. Schott [4] was the first to realise that Born's solution was only valid in the region[1] $x + t > 0$, and that, in fact, the field vanishes in $x + t < 0$. The problem arises at the boundary between these two regions, regarding the application of Maxwell's divergence law ($\nabla \cdot F = -J$, in components $\nabla_\alpha F^{\alpha\beta} = -J^\beta$). The charge in hyperbolic motion presents other complications, such as the question of whether there is a radiation reaction or the implications that this would have for the equivalence principle [5–8].

What strikes us about many of the articles we have consulted is that they use a three-dimensional notation. In this work, we have proposed to calculate the field (and the quantities related to it) always following a covariant tensor notation, taking advantage, as far as possible, of the tools of exterior calculus. In addition to other advantages, the expressions obtained would facilitate the approach of this same problem in curved space-time (in environments with gravity), without forgetting, of course, the difficulties that this would entail.

In Section 3, we briefly review the Liénard–Wiechert solution, in Section 4 we particularise it to hyperbolic motion and analyse its main features (Sections 4.2–4.6). In Section 5, we discuss the region of validity of the solution thus obtained and write the correct expression of the field valid in the whole of the space-time considered. The conclusions are summarised in Section 6 and calculations are detailed in the Appendix A.

## 3. Liénard–Wiechert Solution (Retarded Potentials)

Every electromagnetic field must satisfy Maxwell's equations which, in their covariant version, for external sources located in vacuum, are written simply[2]:

$$dF = 0, \tag{1}$$

$$\nabla \cdot F = -J, \tag{2}$$

where $J$ is the 4-current, which encompasses the charge density $\rho$ and the current density $\vec{j}$ and $F$ is the 2-form of the electromagnetic field: an antisymmetric 2-tensor whose components, referred to an inertial observer, are the electric ($\vec{E}$) and magnetic ($\vec{H}$) fields.

From (1) we already see that $F$ is a closed 2-form and therefore, by the Poincaré lemma, it can be expressed as the exterior differential of a 1-form, in this case, the electromagnetic 4-potential $A$:

$$F = dA. \tag{3}$$

We can now write (2) explicitly in terms of the contravariant components of $A$ and $J$:

$$\Box A^\beta - \partial^\beta(\partial_\alpha A^\alpha) = -J^\beta, \tag{4}$$

where $\Box$ is the Laplace–Beltrami operator in flat space-time ($\Box = \eta^{\mu\nu}\partial_\mu\partial_\nu$). The general solution to this equation, which is obtained starting from the Lorenz (gauge) condition

$\partial_\alpha A^\alpha = 0$ and applying Green's functions (see [9,10]), gives the potential generated by a charged particle[3]:

$$A^\alpha(x) = \int d^4y \, D_r(x - y) \, J^\alpha(y), \tag{5}$$

where $D_r(x - y)$ is the retarded Green's function[4] and $J^\alpha$ the current density produced by the own motion of the point particle with charge $q$:

$$J^\alpha(y) = q \int d\tau \, \dot{x}'^\alpha(\tau) \, \delta(y - x'(\tau)), \tag{6}$$

with $\dot{x}'(\tau)$ the 4-velocity of the charge and $\tau$ its proper time.

If we carry out carefully[5] the two integrals included in (5), we finally obtain the expression of the Liénard–Wiechert potential:

$$A^\alpha(x) = -\left.\frac{q \, \dot{x}'^\alpha}{(x - x')^\beta \dot{x}'_\beta}\right|_{\tau = \tau_1}, \tag{7}$$

where $x'$ is the position 4-vector of the charge and the vertical line indicates that the right-hand side must be evaluated at a proper time $\tau_1$ such that $x - x'(\tau_1)$ is a light-like vector and $t - t'(\tau_1) > 0$, i.e., the potential we measure must have originated at an earlier instant $(t')$ called the retarded time. We will call these conditions, respectively, the first and the second condition of causality.

This potential can be expressed in tensor notation without indices, with $A$ and $u'$ considered either as vector fields or as the respective 1-forms (mapped by the metric):

$$A = -\frac{q \, u'}{\ell \cdot u'}, \tag{8}$$

where $u' = \dot{x}'$ is the charge 4-velocity and $\ell := x - x'$ the light-like vector defined by the position of the charge $x'$ and the point $x$ where we measure the field.

In Appendix A.1, of the appendix we set out the steps to obtain the 2-form of the Liénard–Wiechert field from (8). Using a covariant notation, we obtain:

$$F = -\frac{q}{(\ell \cdot u')^2} \, \ell \wedge \left(a' - \frac{1 + \ell \cdot a'}{\ell \cdot u'} u'\right), \tag{9}$$

where $a'$ is the 4-acceleration of the charge. We can already see that $F$ is a simple 2-form, that is, it results from the exterior product of two 1-forms. For future convenience we can write (9) as:

$$F = -\frac{q}{(\ell \cdot u')^2} \, \ell \wedge \zeta,$$

$$\text{with } \zeta := a' - \frac{1 + \ell \cdot a'}{\ell \cdot u'} u'. \tag{10}$$

The general expression (9) can be rewritten as follows:

$$F = F_{coul} + F_{rad}, \tag{11}$$

with

$$F_{coul} := \frac{q}{(\ell \cdot u')^3} \, \ell \wedge u' \tag{12}$$

the Coulombian part, which only depends on the velocity, and

$$F_{rad} := -\frac{q}{(\ell \cdot u')^2} \, \ell \wedge \left(a' - \frac{\ell \cdot a'}{\ell \cdot u'} u'\right) \tag{13}$$

the radiative part. What is interesting is that the Coulombian part never vanishes (as long as $q \neq 0$), while the radiative part only contributes when the particle is accelerated ($a' \neq 0$).

## 4. Field of the Maxwellian Hyperbolic Charge

*4.1. Intersection of the Light Cone at the Observation Point with the Charge's World Line*

We are ready to calculate the field of the hyperbolic charge using either (8) or directly (9). We will first use the former. Consider that, relative to an inertial observer, the hyperbolic motion of the charge occurs in the $x$ direction, that is, at the intersection between the $(x, y)$ and $(x, z)$ planes. If $\alpha$ is the constant modulus of the charge 4-acceleration ($a'$) and $\beta$ and $\gamma$ any two constants, we can parametrise the motion as a function of its proper time $\tau$ ($x' = x'(\tau)$):

$$t' = \frac{1}{\alpha} sinh(\alpha \tau), \qquad x' = \frac{1}{\alpha} cosh(\alpha \tau), \qquad y' = \beta, \qquad z' = \gamma. \tag{14}$$

By eliminating the proper time $\tau$, we can describe this hyperbolic motion only in terms of the coordinates measured by the inertial observer:

$$(x')^2 - (t')^2 = \frac{1}{\alpha^2}, \qquad y' = \beta, \qquad z' = \gamma. \tag{15}$$

We want to express the variables appearing in (8) as a function of x in order to calculate the exterior differential of the potential (with respect to x) and thus obtain the field $F$. To do this, it is sufficient to solve the system of equations composed of Equations[6] (15) and the equation of a light cone centred at x':

$$-(t - t')^2 + (x - x')^2 + (y - y')^2 + (z - z')^2 = 0. \tag{16}$$

If we analyse Figure 1, we can obtain valuable information about the field we are looking for, even before we try to solve this system of equations. Since the particle's trajectory is confined to the region defined by $|t| + x > 0$ (region I in Figure 1), this system of equations is only solvable for $x$ in regions I, II, and IV, so we conclude that the field vanishes in region III ($|t| + x < 0$).

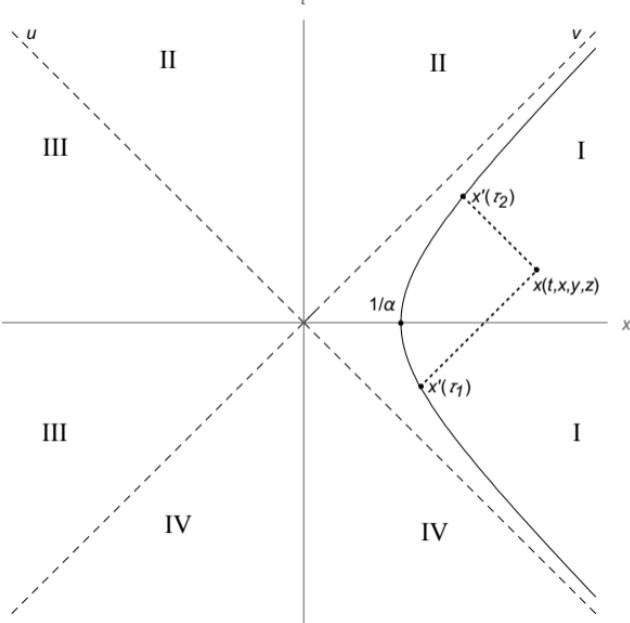

**Figure 1.** Charge trajectory and division of space-time into 4 regions (I, II, III, and IV). The two represented instants of the trajectory ($\tau_1$ and $\tau_2$, with $\tau_2 > \tau_1$) are on the light cone with vertex at x. However, only at the instant $\tau_1$ can a measurable field in $x$ be created. Two spatial dimensions have been omitted in the figure.

Moreover, being a quadratic system, we expect to obtain two possible solutions, $x'(\tau_1)$ and $x'(\tau_2)$, for a given x. One of them is interpreted as the one in which x lies in the past light cone of x' ($x'(\tau_2)$) and, therefore, any light signal emitted in x' would not affect the field measured in x. The other solution describes the reverse situation: here x lies in the future light cone of x' ($x'(\tau_1)$). We must choose the latter, since it is the one that satisfies the second causality condition ($t > t'$). Still looking at Figure 1, we see that in region II and region IV there can only be one possible solution for each $x$. Moreover, in region IV the solution does not satisfy the second causality condition, so the field will be equal to zero also in that region. Therefore, at this point we can already say that the field we are going to obtain will have a nonzero value only in regions I and II, that is, for $x + t > 0$. Later we will discuss what happens at the boundary between regions I and IV and between II and III ($x + t = 0$).

In Appendix A.2, we write the two solutions ($x' = x'(x)$) of the system of Equations (15) and (16), with $x' > 0$, and select the one that satisfies $t' < t$ (A11). In this way we obtain the correct expression of the light-like vector $\ell$ as a function of $t, x, y$ and $z$ (A14). Now we only need the 4-velocity u' of the charge. Considering that $u' = \dot{x}'$, based on Equation (14), we obtain:

$$\dot{t}' = \alpha \, x', \qquad \dot{x}' = \alpha \, t', \qquad \dot{y}' = 0, \qquad \dot{z}' = 0. \tag{17}$$

Now we also have u' as a function of $t, x, y$ and $z$ (A15) and we can write (8).

### 4.2. The 1-Form A Related with the Vector Potential

In Appendix A.3 we explicitly write (8). The expression we obtain invites us to make the following change of coordinate system:

$$u = t - x, \qquad v = t + x, \qquad \rho^2 = (y - \beta)^2 + (z - \gamma)^2, \qquad \chi = \arctan\left(\frac{z - \gamma}{y - \beta}\right). \tag{18}$$

From now on, we will work in coordinates[7] $(u, v, \rho, \chi)$, with the Minkowski metric expressed as:

$$\eta = -\frac{1}{2} du \widetilde{\otimes} dv + d\rho \otimes d\rho + \rho^2 d\chi \otimes d\chi, \tag{19}$$

where we have used $du \widetilde{\otimes} dv = du \otimes dv + dv \otimes du$.

Later, in order to calculate the divergence of $F$, it will be useful to know that

$$\sqrt{|det\eta|} = \frac{\rho}{2}. \tag{20}$$

This change to a cylindrical coordinate system[8] is due to the azimuthal symmetry of the field, as we will see.

In these coordinates, the potential, written as 1-form, reads:

$$A = \frac{q}{2f^{1/2}} \left\{ \frac{1}{u}[1 + \alpha^2(\rho^2 - uv) - f^{1/2}]du - \frac{1}{v}[1 + \alpha^2(\rho^2 - uv) + f^{1/2}]dv \right\} \theta(v), \tag{21}$$

with[9]

$$f := \alpha^4(\rho^2 - uv)^2 + 2\alpha^2(\rho^2 + uv) + 1. \tag{22}$$

In (21) we have added the Heaviside step function[10] $\theta(v)$ to limit this result to its region of validity (I and II), the potential vanishing in the other regions. However, as it is written, the potential is not valid in the light-like 3-plane $v = 0$. In Section 5, we will obtain the complete field by adding a term to (25).

We see that this potential is singular when the function $f$ goes to zero. This occurs precisely when we are on the point charge, i.e., when $u\,v = -1/\alpha^2$ and $\rho = 0$. In addition, the potential appears to exhibit another singularity when $u \to 0$ with $v > 0$. However, we verify that when $u \to 0$, the numerator $[1 + \alpha^2(\rho^2 - uv) - f^{1/2}] \to 0$, resulting in a finite

limit. On the other hand, when $v \to 0^+$ the potential is singular and here we see that in this border things get complicated. We will discuss this in Section 5.

*4.3. The Field F and Its Dual*

To calculate the electromagnetic field we obtain the exterior differential of the potential (21), separating the step function so that $A := \widetilde{A}\,\theta(v)$:

$$F = dA = d\widetilde{A}\,\theta(v) + \widetilde{A} \wedge d\theta(v). \tag{23}$$

Let us look at the second term of (23). Taking into account that $\widetilde{A}$ has only $u$ and $v$ components and that the 1-form[11] $d\theta(v) = \delta(v)\,dv$, where $\delta(v)$ is the Dirac distribution, the only component that will survive the exterior product will be:

$$\widetilde{A} \wedge d\theta(v) = \frac{q}{2f^{1/2}u}[1 + \alpha^2(\rho^2 - uv) - f^{1/2}]\,\delta(v)\,du \wedge dv. \tag{24}$$

The presence of the Dirac delta selects, in the sense of a distribution, the value $v = 0$. It is easy to check that then this term vanishes[12]. Therefore, to obtain the field, we only have to calculate the exterior differential of $\widetilde{A}$ and then add the step function.

In this way, we obtain the 2-form of the electromagnetic field of a point charge in hyperbolic motion for the region $v \neq 0$[13]:

$$F = \frac{2q\alpha^2}{f^{3/2}}[(1 + \alpha^2(\rho^2 + uv))du \wedge dv - 2\alpha^2 v\rho\,du \wedge d\rho + 2\alpha^2 u\rho\,dv \wedge d\rho]\,\theta(v). \tag{25}$$

To continue, it is convenient to calculate the (Hodge) dual of $F$:

$$*F = -\frac{4q\alpha^2\rho}{f^{3/2}}[\alpha^2\rho vs.\,du \wedge d\chi + \alpha^2\rho u\,dv \wedge d\chi - (1 + \alpha^2(\rho^2 + uv))d\rho \wedge d\chi]\,\theta(v). \tag{26}$$

Now we obtain the electric field $e$ and the magnetic field $h$ with respect to the inertial observer $\mathbf{u} = -\frac{1}{2}(du + dv)$:

$$e = -i(\mathbf{u})F = -\frac{2q\alpha^2}{f^{3/2}}[(1 + \alpha^2(\rho^2 + uv))(du - dv) - 2\alpha^2\rho(u - v)d\rho]\,\theta(v), \tag{27}$$

$$h = -i(\mathbf{u}) * F = -\frac{4q\alpha^4(u + v)}{f^{3/2}}\rho^2\,d\chi\,\theta(v). \tag{28}$$

Now we can write their non-vanishing contravariant components:

$$e^x = \frac{4q\alpha^2}{f^{3/2}}\left(1 + \alpha^2(\rho^2 + t^2 - x^2)\right)\theta(t + x),$$

$$e^\rho = -\frac{8q\alpha^4}{f^{3/2}}x\rho\,\theta(t + x), \tag{29}$$

$$h^\chi = -\frac{8q\alpha^4}{f^{3/2}}t\,\theta(t + x).$$

With these results we can check that the algebraic decomposition of $F$ (and $*F$) in terms of these relative quantities is satisfied:

$$F = \mathbf{u} \wedge e - *(\mathbf{u} \wedge h), \qquad *F = \mathbf{u} \wedge h + *(\mathbf{u} \wedge e). \tag{30}$$

Furthermore, we calculate the scalar invariants of the field:

$$f := \frac{1}{2}trF^2 = e^2 - h^2 = \frac{16q^2\alpha^4}{f^2}\,\theta(v),$$

$$\widetilde{f} := \frac{1}{2}tr(F \times *F) = 2\,e \cdot h = 0. \tag{31}$$

It is therefore a regular field. This result is also evident from the general expression of the Liénard–Wiechert field (9). Recall that a regular field is one with at least one nonzero invariant, which can be generally decomposed as:

$$F = \mu\, n \wedge k + \nu * (n \wedge k), \tag{32}$$

with $n$ and $k$ the principal directions of $F$, which are necessarily light-like. These are the eigenvectors of $F$ with respective eigenvalues $\mu$ and $-\mu$ (they are also the eigenvectors of $*F$ with respective eigenvalues $-\nu$ and $\nu$). We take them on the same half of the light cone, with $n \cdot k = -1$. In our case, since the second invariant is equal to zero (31), necessarily[14] $\nu = 0$ and (32) reads:

$$F = \mu\, n \wedge k, \tag{33}$$

where now $\mu = f$, with $f$ the nonzero invariant of the field $F$ given in (31).

Looking at the expression (10), since $\ell$ is light-like by definition, we might be tempted to think that $\zeta$ is also light-like and that $\ell$ and $\zeta$ are the two principal directions of $F$. However, this is not the case, since $\zeta$ is not light-like. However, it is striking that $\ell \cdot \zeta = -1$, as required for the two principal directions of a regular field[15]. It is easy to check that the vector defined as

$$k = \zeta + \frac{1}{2}\zeta^2 \ell \tag{34}$$

is light-like with $k \cdot \ell = -1$ and, furthermore, that

$$\ell \wedge k = \ell \wedge \zeta, \tag{35}$$

as it appears in the general expression of the Liénard–Wiechert field (10).

Now we have the two principal directions, $\ell$ and $k$, of our regular field, also called a non-radiative field alluding to the set of observers, those located on the 2-plane $(\ell, k)$, for whom the Poynting vector is zero and therefore do not see an energy flux.

*4.4. The Stress-Energy Tensor T*

Let us now calculate the stress-energy tensor of the electromagnetic field, defined as

$$T = -F^2 + \frac{1}{4} tr F^2 \eta. \tag{36}$$

In the present case, using the coordinate system (18), we obtain (expressed as a contravariant 2-tensor):

$$
\begin{aligned}
T = \frac{8q^2\alpha^4}{f^3} \Big\{ & 8\alpha^4 \rho^2 \left( u^2\, \partial_u \otimes \partial_u + v^2\, \partial_v \otimes \partial_v \right) + 2(1 + \alpha^2(uv + \rho^2))^2\, \partial_u \widetilde{\otimes} \partial_v + \\
& + 4\alpha^2 \rho \left( 1 + \alpha^2(uv + \rho^2) \right)\left( u\, \partial_u \widetilde{\otimes} \partial_\rho + v\, \partial_v \widetilde{\otimes} \partial_\rho \right) + \\
& + \frac{f}{\rho^2}\, \partial_\chi \otimes \partial_\chi + \left( f + 8\alpha^4 uv\rho^2 \right) \partial_\rho \otimes \partial_\rho \Big\} \theta(v).
\end{aligned}
\tag{37}
$$

From the stress-energy tensor, we calculate the quantities relative to the inertial observer u, the electromagnetic energy $\widetilde{\rho}$ and the Poynting vector $s$:

$$
\begin{aligned}
\widetilde{\rho} = T(\mathsf{u}, \mathsf{u}) &= \frac{1}{2}(e^2 + h^2) \\
&= \frac{8q^2\alpha^4}{f^3}\Big\{ (1 + \alpha^2(\rho + uv))^2 + 2\alpha^4 \rho^2(u^2 + v^2) \Big\}\theta(v),
\end{aligned}
\tag{38}
$$

$$
\begin{aligned}
s &= -[i(\mathsf{u})T]_\perp \\
&= \frac{16q^2\alpha^6(u + v)}{f^3}\Big\{ \alpha^2(u - v)\rho^2(\partial_u - \partial_v) - (1 + \alpha^2(uv + \rho^2))\rho\, \partial_\rho \Big\}\theta(v),
\end{aligned}
\tag{39}
$$

where $[\;]_\perp$ denotes the orthogonal projection onto the observer's reference space.

Here we read the contravariant components of $s$:

$$s^x = \frac{128\,q^2\alpha^8 tx\rho^2}{f^3}\theta(v),$$

$$s^\rho = -\frac{32q^2\alpha^6 t\rho}{f^3}(1+\alpha^2(t^2-x^2+\rho^2))\theta(v). \tag{40}$$

Here we see that at $t = 0$ the Poynting vector is zero. This is discussed in the next section.

### 4.5. Characteristics of the Electric and Magnetic Field

From the expressions (27) and (28) we see that both the electric field $e$ and the magnetic field $h$ have azimuthal symmetry (their components do not depend on $\chi$). Moreover, the $h$ field has only an azimuthal component, while the $e$ field does not, so that $e \cdot h = 0$ at each instant $t$.

Figure 2 shows the lines of the field $e$ on the plane $(x, \rho)$ at $t = 0$ and at $t > 0$. We see how the lines form circles centred on the $\rho$ axis that pass through the position of the charge. To calculate the centre of these circles we must find the family of functions $\rho_i = \rho_i(x)$ representing the field lines, solving for each $t$ the following differential equation:

$$\frac{d\rho_i}{dx} = \frac{e^\rho}{e^x} = \frac{-2\alpha^2\rho_i\,x}{1+\alpha^2(\rho_i^2+t^2-x^2)}, \tag{41}$$

where we have used (29) for the contravariant components of the electric field, $e^\rho$ and $e^x$.

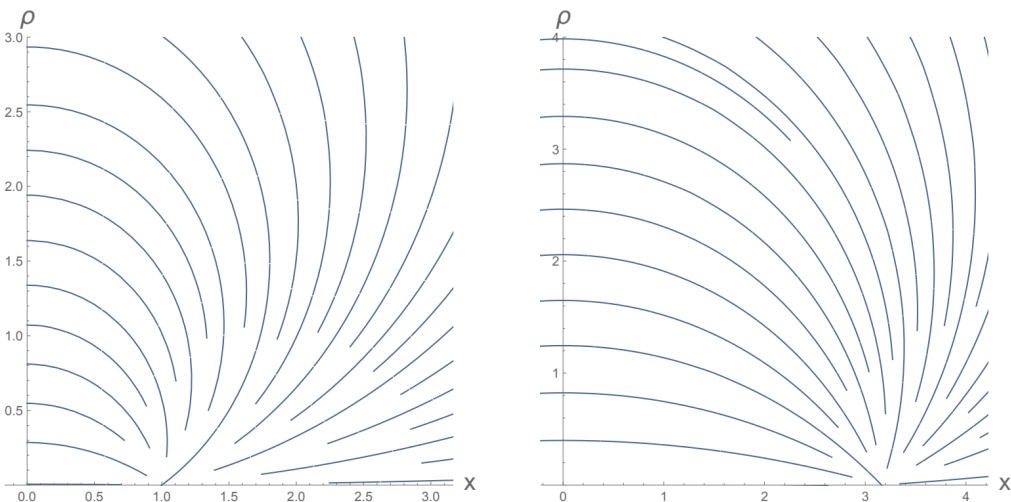

**Figure 2.** Electric field lines at $t = 0$ (**left**) and $t' > 0$ (**right**). At $t = 0$ the charge is closer to the observer (here at $x = 1$). The cutting of the lines on the $\rho$ axis is due to the fact that at $t = 0$ the field vanishes for $x < 0$ (we will discuss the field at $x + t = 0$ in the next sections). In the image on the right that cut occurs at $x < -t'$. The third space-like coordinate $\chi$ has been omitted.

The solution to this differential equation reflects the circular shape, centred on the $\rho$-axis, of these field lines:

$$(\rho_i - C_i)^2 + x^2 = \frac{1}{\alpha^2} + t^2 + C_i^2, \tag{42}$$

where each solution $\rho_i$ is determined by the value of $C_i \in \mathbb{R}^+$, the integration constant.

The discontinuity in the light-like 3-plane $v = 0$ of the $F$ field (25) causes an interruption of the electric field lines on the $\rho$ axis, as seen in Figure 2 for $t = 0$ (*left*). The interesting part is that, if we approach that boundary from $v > 0$ ($v \to 0^+$), the electric field has a finite

(nonzero) value. Let us see it explicitly for $t = 0$. In that case, by making $t = 0$ and $x \to 0^+$ in (29), we obtain the nonzero component:

$$
\begin{aligned}
e^x &= \frac{4q\alpha^2}{f^{3/2}_{t=0, x\to 0^+}} \left(1 + \alpha^2 \rho^2\right) \\
&= \frac{4q\alpha^2}{(1 + \alpha^2 \rho^2)^2}.
\end{aligned}
\tag{43}
$$

Therefore, on the right-hand side of the boundary $t + x = 0$, the electric field has a nonzero component, the $x$ component. We can integrate the flow of these field lines through the (differential) circular crown $2\pi \rho d\rho$ centred on the $x$ axis, whose normal vector $n$ is directed in the direction of this axis, covering the whole space. In this case we have:

$$
8\pi q \alpha^2 \int_0^\infty \frac{\rho}{(1 + \alpha^2 \rho^2)^2} d\rho = 4\pi q
\tag{44}
$$

This is the result we expected[16] according to Gauss' law of divergence. Since we are dealing with a point charge, the volume integral of the charge distribution in the half-space $x > 0$ is equal to the charge $q$ of the particle. On the other hand, the flux of the electric field through the edge $x = 0$ of this half-space, is the integral that we have calculated in (44). We can say that the field lines do not disappear at the discontinuity $v = 0$, but remain within the region $v > 0$. In Section 5, we will add a term to (25) so that the $F$ field satisfies Maxwell's divergence law $\nabla \cdot F = -J$ in accordance with this result.

Another interesting property is that the magnetic field $h$ cancels out at the instant $t = 0$ ($\Leftrightarrow u + v = 0$)[17]. At this instant the Poynting vector $s$ (40) is also zero. This property is what sparked the debate about whether a uniformly accelerated charge radiates energy or not. Some authors, W. Pauli [12] among them[18], maintained that the hyperbolic charge did not radiate energy, arguing that by successive changes of the inertial system one could continuously place oneself in an inertial system so that $s = 0$ at each instant. Other authors such as H. Bondi and T. Gold [6] and T. Fulton and F. Rohrlich [13] refuted this argument years later.

In this regard we will say that, from a more algebraic perspective, we are dealing with a regular field, so that for an observer located on the 2-plane formed by the two principal directions $\{\ell, k\}$, the Poynting vector is always zero. It is easy to verify that, in our case, at instant $t = 0$, the 2-plane formed by $\{\ell, k\}$ is oriented in such a way that our observer u is precisely on it and, therefore, sees no energy flow at that instant. Therefore, the question of whether the field of a charge in hyperbolic motion radiates energy or not could be reformulated in the sense of whether a regular field such as this one radiates energy or not.

*4.6. The Function f*

We have said that the function $f$ (22) is interesting and it is so for several reasons. This function appears in all our expressions, starting with the solution $x' = x'(x)$ (A11) to the system of Equations (15) and (16) and, from there, in all the quantities we have obtained, always in the denominator and raised to different powers. As we have pointed out, it vanishes when we place ourselves on the charge's world line, giving rise, as expected, to singular expressions in that case. However, the property that catches our attention is that the nonzero scalar invariant of the field, $f$, is proportional precisely to $f^{-2}$ (see (31)).

To study this function we will analyse the causal character of the level 3-surfaces $f = constant$ according to the values of $uv$ and $\rho^2$. In the following analysis we must not forget that, although the function $f$ is defined for all $v$, our field vanishes at $v < 0$.

We know that $df$ is the 1-form giving an orthogonal direction to the surfaces $f = constant$:

$$
\begin{aligned}
df =& 2\alpha^2 (1 - \alpha^2 v(\rho^2 - uv))\, du + 2\alpha^2 (1 - \alpha^2 u(\rho^2 - uv))\, dv \\
& + 4\alpha^2 \rho (1 + \alpha^2 (\rho^2 - uv))\, d\rho.
\end{aligned}
\tag{45}
$$

To obtain the causal character we calculate the square of $df$:

$$\begin{aligned}
\eta(df, df) &= 16\alpha^4(\rho^2 - uv)(\alpha^4(\rho^2 - uv)^2 + 2\alpha^2(\rho^2 + uv) + 1) \\
&= 16\alpha^4(\rho^2 - uv)f.
\end{aligned} \tag{46}$$

Furthermore, we check the sign of (46), so that if $\eta(df, df) = 0$ the 3-surfaces are light-like, if $\eta(df, df) > 0$ they are time-like and if $\eta(df, df) < 0$ they are space-like. Looking at (22) we realize that the function $f$ can be written as the following sum of squares:

$$f = \left(\alpha^2(uv - \rho^2) + 1\right)^2 + 4\alpha^2\rho^2. \tag{47}$$

Therefore, $f \geq 0$ for all $uv$ and $\rho^2$, so the sign of $\eta(df, df)$ will be determined by the sign of $\rho^2 - uv$.

Table 1 summarises the causal character of the 3-level surfaces $f = constant$. In Figure 3 we try to visualise these regions (see Appendix A.4 for more details).

**Table 1.** Causal character of the level surfaces $f = constant$ according to the values of $uv$ and $\rho^2$.

|  | $uv = 0$ | $uv > 0$ | $uv = \rho^2$ | $uv < 0$ |
|---|---|---|---|---|
| $\rho^2 = 0$ |  | Space-like |  | Time-like |
| $\rho^2 > 0$ | Time-like |  | Light-like |  |
| $\rho^2 > uv$ |  | Time-like |  |  |
| $\rho^2 < uv$ |  | Space-like |  |  |

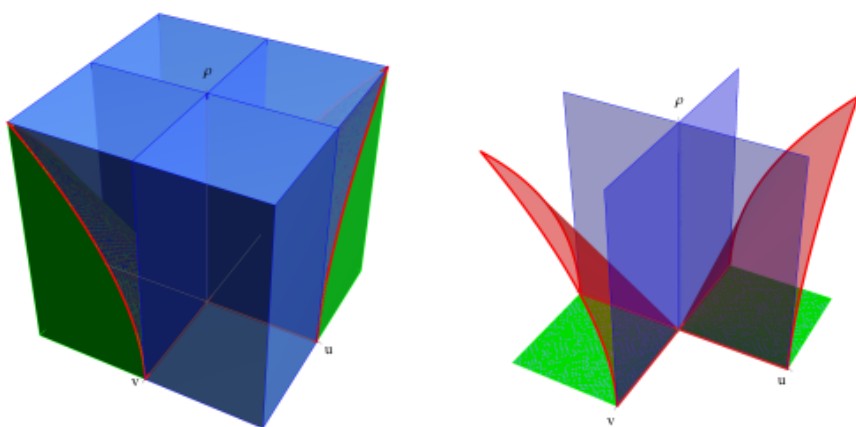

**Figure 3.** Trying to visualise the regions of Table 1 (left). Three of these regions (those with coloured initial letter) are shown on the right. The light cone corresponds to $uv = \rho^2$. Colour coding: Time-like Space-like Light-like.

## 5. Region of Validity of the Field

As already anticipated, the expression of the field (25) is not valid on the light-like 3-plane $v = 0$. The reason, as we shall see, is that at that boundary, where the field goes from having a finite value to being zero, Maxwell's equations are not satisfied. However, why are they not satisfied if precisely those equations have been our starting point? As explained in [14][19], the Green's functions method used to obtain the potential, which we have briefly described here, fails when the trajectory tends asymptotically to a light-like trajectory, as occurs in our case in $t \to -\infty$. If we look at the expression (8), we see that the denominator $(\ell \cdot u')$ vanishes when the charge velocity four-vector $u'$ is light-like (as it occurs in $t \to -\infty$) and collinear with the light-like vector of the point of observation $\ell$ (as it can occur when we measure at a point on the 3-surface $v = 0$), giving rise to a

singularity. As explained by Bondi and Gold [6], the problem is that the contributions to the field from these distant regions do not attenuate fast enough, leading to a divergent term at that boundary.

Let us see explicitly that (25) does not satisfy Maxwell's equations in the region $v = 0$. The presence of the Heaviside *distribution* in (25) already tells us that we are now dealing with tensor distributions, so we must reinterpret Maxwell's equations in the sense of distributions[20].

When we are not on the charge's trajectory ($\rho \neq 0$ and $u$ *vs.* $\neq -\frac{1}{\alpha^2}$) and, therefore, $J = 0$, Maxwell's second Equation (2) in component notation says:

$$\nabla_\mu F^{\mu\nu} = 0. \tag{48}$$

For the following calculations it is convenient to write the contravariant components of $F$:

$$F^{uv} = -F^{vu} = -\frac{8q\alpha^2}{f^{3/2}}(1 + \alpha^2(\rho^2 + uv))\,\theta(v),$$

$$F^{u\rho} = -F^{\rho u} = -\frac{8q\alpha^4 u\rho}{f^{3/2}}\,\theta(v), \tag{49}$$

$$F^{v\rho} = -F^{\rho v} = \frac{8q\alpha^4 v\rho}{f^{3/2}}\,\theta(v).$$

If we separate the function $\theta(v)$ from the components (49) such that:

$$F^{\mu\nu} := \widetilde{F}^{\mu\nu}\,\theta(v), \tag{50}$$

we have:

$$\nabla_\mu F^{\mu\nu} = \nabla_\mu(\widetilde{F}^{\mu\nu})\,\theta(v) + \widetilde{F}^{\mu\nu}(\nabla_\mu\theta(v)). \tag{51}$$

To calculate the first term of (51), when working with curvilinear coordinates, we must take into account that

$$\nabla_\mu(\widetilde{F}^{\mu\nu}) = \frac{1}{\sqrt{|det\eta|}}\partial_\mu(\sqrt{|det\eta|}\,\widetilde{F}^{\mu\nu}), \tag{52}$$

where the factor with the metric determinant has already been calculated in (20).

With this we verify that $\nabla_\mu(\widetilde{F}^{\mu\nu}) = 0$, i.e., in the region $v > 0$, where $F = \widetilde{F}$, $\nabla \cdot F = 0$ is satisfied. This does not surprise us because we know that our solution, as written above, is valid in $v > 0$ and, therefore, satisfies Maxwell's equations in that region.

To calculate the second term of (51) we define:

$$\bar{J}^\nu := \widetilde{F}^{\mu\nu}(\nabla_\mu\theta(v)). \tag{53}$$

Bearing in mind that $\nabla_\mu\theta(v) = \delta_\mu^v\,\delta(v)$, we obtain the following nonzero components of $\bar{J}$:

$$\bar{J}^u = -\frac{8q\alpha^2}{f^{3/2}}(1 + \alpha^2(\rho^2 + uv))\,\delta(v),$$

$$\bar{J}^\rho = \frac{8q\alpha^4 v\rho}{f^{3/2}}\,\delta(v). \tag{54}$$

These expressions can be simplified by taking into account that the Dirac delta selects, in the sense of a distribution, the value $v = 0$, so that the only nonzero component of $\bar{J}$ is:

$$\bar{J}^u = -\frac{8q\alpha^2}{(1 + \alpha^2\rho^2)^2}\,\delta(v). \tag{55}$$

This term, which we can see as an effective current density at the 3-surface $v = 0$, is what prevents the second Maxwell equation from being satisfied at that boundary. Therefore, to obtain the correct expression of the field we need to add a term whose

divergence cancels this effective current and only comes into play at $v = 0$. Based on Boulware's article [7], we prove that this term, written as a contravariant 2-tensor, is:

$$\overline{F} = -\frac{4q\alpha^2\rho}{1 + \alpha^2\rho^2}\, \delta(v)\left[\partial_u \otimes \partial_\rho - \partial_\rho \otimes \partial_u\right]. \tag{56}$$

Let us see explicitly that this term is the one we are looking for. Since $\overline{F}$ only depends on $\rho$ and $v$, the only nonzero component of $\nabla_\mu \overline{F}^{\mu\nu}$ is the $u$-component:

$$[\nabla \cdot \overline{F}]^u = \frac{1}{\rho}\partial_\rho(\rho\overline{F}^{\rho u}) = \frac{1}{\rho}\partial_\rho\left[\frac{4q\alpha^2\rho^2}{1 + \alpha^2\rho^2}\right]\delta(v) + \frac{4q\alpha^2}{1 + \alpha^2\rho^2}\partial_\rho[\delta(v)]. \tag{57}$$

The second summand of (57) is zero under conmutation of the partial derivative[21] and therefore:

$$[\nabla \cdot \overline{F}]^u = \frac{8q\alpha^2}{(1 + \alpha^2\rho^2)^2}\delta(v) = -\overline{J}^u. \tag{58}$$

We can now write the components of the field $F = \widetilde{F}\,\theta(v) + \overline{F}$ of the hyperbolic charge which satisfies Maxwell's equations also in the half-space $v \geq 0$:

$$F^{uv} = -F^{vu} = -\frac{8q\alpha^2}{f^{3/2}}\left(1 + \alpha^2(\rho^2 + uv)\right)\theta(v),$$

$$F^{u\rho} = -F^{\rho u} = -4q\alpha^2\rho\left[\frac{2\alpha^2 u}{f^{3/2}}\,\theta(v) + \frac{\delta(v)}{1 + \alpha^2\rho^2}\right], \tag{59}$$

$$F^{v\rho} = -F^{\rho v} = \frac{8q\alpha^4 v\rho}{f^{3/2}}\,\theta(v).$$

With the correct field, we recalculate the electric field $e$ and obtain its contravariant components $e^x$ and $e^\rho$:

$$e^x = \frac{4q\alpha^2}{f^{3/2}}\left(1 + \alpha^2(\rho^2 + t^2 - x^2)\right)\theta(t + x),$$

$$e^\rho = -\frac{8q\alpha^4}{f^{3/2}}x\rho\,\theta(t + x) + \frac{2q\alpha^2\rho}{1 + \alpha^2\rho^2}\,\delta(t + x). \tag{60}$$

For the magnetic field we obtain:

$$h^\chi = -\frac{8q\alpha^4}{f^{3/2}}t\,\theta(t + x) - \frac{2q\alpha^2}{1 + \alpha^2\rho^2}\,\delta(t + x). \tag{61}$$

We see that the $x$-component of the electric field does not change. The term that we have added to $e^\rho$ and $h^\chi$ only intervenes at $x + t = 0$ and comes from the effective current $\overline{J}$.

## 6. Conclusions and Perspectives

In this article we have analysed the electromagnetic field produced by a massive point charge in hyperbolic motion in Minkowski space-time. We have started from the Liénard–Wiechert potential as a solution to Maxwell's equations for the electromagnetic field generated by a point charge in arbitrary motion. We have particularised this potential to hyperbolic motion and obtained, in covariant notation, the quantities that characterise this field.

We have analysed the electric and magnetic field (referred to an inertial observer) of the charge in hyperbolic motion, drawn the electric field lines at two instants and verified that Gauss' law of divergence is satisfied at the critical boundary $\{x = 0, t = 0\}$, concluding that the field lines do not disappear at that boundary, but remain within the region $x + t > 0$. We have briefly addressed the problem of whether a uniformly accelerated charge radiates

energy or not: for an observer on the 2-plane formed by the principal directions of a regular electromagnetic field, such as the Liénard–Wiechert field, the Poynting vector is always zero. At $t = 0$ our observer is precisely on that plane and sees no energy flux. Furthermore, we have noticed that the non-vanishing scalar invariant of the field, $f$, is proportional to the function $f = f(u, v, \rho)$ which appears in the denominator of all quantities that we have calculated (the vector potential, the field, the stress-energy tensor, ...). We have analysed this function in detail: we have studied the causal character of the level surfaces $f = constant$ and tried to visualize these causal regions, as well as the level surfaces.

We have realised that, since there is a discontinuity in the field of the charge in hyperbolic motion, the Liénard–Wiechert solution only gave us the field in a certain region, precisely the one that excludes that discontinuity. Reinterpreting Maxwell's equations in the sense of distributions we have found a solution that satisfies these equations in all the space-time considered.

The study we have carried out allows us to pose this same problem in curved space-time. We believe that we will be able to take advantage of the covariant nature of the expressions obtained. However, there will be concepts that we will have to define carefully, such as the light-like vector $\ell$ defined by the position of the charge x′ and the point x where we measure the field—to mention one of them.

The techniques we have learned in this analysis, such as the approach of the intersection of the light cone with vertex at $x$ with the trajectory of the charge (the system of equations composed of (15) and (16)), can be used to enter the field of relativistic positioning [16,17]. For example, we could consider the case of four hyperbolic emitters in different time-like planes and solve the resulting system of equations to obtain an expression of the coordinates (with respect to an inertial observer) as a function of the proper time of each of these emitters.

**Author Contributions:** Both authors contributed equally to develop the idea of the manuscript. All authors have read and agreed to the published version of the manuscript.

**Funding:** This work was carried out within the Spanish Ministerio de Ciencia, Innovación y Universidades Project PID2019-109753GB-C21/AEI/10.13039/501100011033.

**Data Availability Statement:** Not applicable.

**Acknowledgments:** We would like to thank the /Universe/ Editorial Office at MDPI, for the invitation to submit this manuscript free of charge.

**Conflicts of Interest:** The authors declare no conflict of interest.

## Appendix A

In this appendix we include those expressions and calculations which, due to their extension, should be placed in a separate section so as not to hinder the reading of the manuscript.

The order of the sections in this appendix is determined by the main text itself.

*Appendix A.1. Deduction of the Liénard–Wiechert Field from the Potential in Covariant Notation*

To obtain the *F* field from (8):

$$F = dA = \frac{q}{(\ell \cdot \mathbf{u}')^2} \, d(\ell \cdot \mathbf{u}') \wedge \mathbf{u}' - \frac{q}{\ell \cdot \mathbf{u}'} \, d\mathbf{u}'. \tag{A1}$$

Let us first calculate $d(\ell \cdot \mathbf{u}')$ in components:

$$[d(\ell \cdot \mathbf{u}')]_\alpha = (\partial_\alpha \ell^\mu) \mathbf{u}'_\mu + \ell^\mu (\partial_\alpha \mathbf{u}'_\mu). \tag{A2}$$

Let us see if we can express the partial derivatives that appear in (A2) in terms of other more practical quantities.

From the definition of $\ell = \mathrm{x} - \mathrm{x}'$:

$$\partial_\alpha \ell^\mu = \delta^\mu_\alpha - \partial_\alpha \mathrm{x}'^\mu. \tag{A3}$$

Since $\ell$ is light-like:

$$0 = \ell^2 = \eta_{\mu\nu}(\mathrm{x}^\mu - \mathrm{x}'(\tau)^\mu)(\mathrm{x}^\nu - \mathrm{x}'(\tau)^\nu),$$

differentiating:

$$0 = 2\eta_{\mu\nu}(d\mathrm{x}^\mu - \tfrac{d\mathrm{x}'^\mu}{d\tau}d\tau)(\mathrm{x}^\nu - \mathrm{x}'(\tau)^\nu)$$
$$\Leftrightarrow 0 = \eta_{\mu\nu}(d\mathrm{x}^\mu - \mathrm{u}'^\mu d\tau)\ell^\nu = \ell_\mu d\mathrm{x}^\mu - (\ell \cdot \mathrm{u}')d\tau$$
$$\Rightarrow d\tau = \frac{\ell}{\ell \cdot \mathrm{u}'}. \tag{A4}$$

Now:

$$d\mathrm{x}'^\mu = \mathrm{u}'^\mu d\tau = \mathrm{u}'^\mu \frac{\ell}{\ell \cdot \mathrm{u}'}$$
$$\Leftrightarrow \partial_\alpha \mathrm{x}'^\mu = \frac{\mathrm{u}'^\mu \ell_\alpha}{\ell \cdot \mathrm{u}'}. \tag{A5}$$

Therefore, (A3) reads:

$$\partial_\alpha \ell^\mu = \delta^\mu_\alpha - \frac{\mathrm{u}'^\mu \ell_\alpha}{\ell \cdot \mathrm{u}'}. \tag{A6}$$

Let us continue with the second summand of (A2):

$$d\mathrm{u}'_\mu = \mathrm{a}'_\mu d\tau = \mathrm{a}'_\mu \frac{\ell}{\ell \cdot \mathrm{u}'}$$
$$\Leftrightarrow \partial_\alpha \mathrm{u}'_\mu = \frac{\mathrm{a}'_\mu \ell_\alpha}{\ell \cdot \mathrm{u}'}, \tag{A7}$$

where $\mathrm{a}' = \acute{\mathrm{u}}' = \ddot{\mathrm{x}}'$ is the 4-acceleration of the charge.

Now we can rewrite (A2):

$$[d(\ell \cdot \mathrm{u}')]_\alpha = \left[\delta^\mu_\alpha - \frac{\mathrm{u}'^\mu \ell_\alpha}{\ell \cdot \mathrm{u}'}\right]\mathrm{u}'_\mu + \ell^\mu \left[\frac{\ell_\alpha \mathrm{a}'_\mu}{\ell \cdot \mathrm{u}'}\right]. \tag{A8}$$

Since $\mathrm{u}'^2 = -1$, we can simplify (A8) and write it as 1-form:

$$d(\ell \cdot \mathrm{u}') = \mathrm{u}' + \frac{1 + \ell \cdot \mathrm{a}'}{\ell \cdot \mathrm{u}'}\ell. \tag{A9}$$

It is now easy to calculate $d\mathrm{u}'$ as it appears in (A1) using the definition of the exterior derivative, $[d\mathrm{u}']_{\alpha\beta} = \partial_\alpha \mathrm{u}'_\beta - \partial_\beta \mathrm{u}'_\alpha$, and (A7):

$$d\mathrm{u}' = \frac{\ell \wedge \mathrm{a}'}{\ell \cdot \mathrm{u}'}. \tag{A10}$$

Finally, we substitute (A9) and (A10) into (A1) to obtain the Liénard–Wiechert field:

$$F = -\frac{q}{(\ell \cdot \mathrm{u}')^2} \ell \wedge \left(\mathrm{a}' - \frac{1 + \ell \cdot \mathrm{a}'}{\ell \cdot \mathrm{u}'}\mathrm{u}'\right).$$

*Appendix A.2. Expression of the Retarded Coordinates $x'$ as a Function of the Point $x$ Where the Field Is Measured*

The system of equations consisting of (15) and (16) has the following two solutions (which differ in the sign of the square root of $f$):

$$t'_1 = -\frac{\alpha^2 t(\rho^2 - t^2 + x^2) + t - xf^{\frac{1}{2}}}{2\alpha^2(t^2 - x^2)},$$

$$x'_1 = \frac{\alpha^2 x(\rho^2 - t^2 + x^2) + x - tf^{\frac{1}{2}}}{2\alpha^2(x^2 - t^2)}, \tag{A11}$$

$$t'_2 = -\frac{\alpha^2 t(\rho^2 - t^2 + x^2) + t + xf^{\frac{1}{2}}}{2\alpha^2(t^2 - x^2)},$$

$$x'_2 = \frac{\alpha^2 x(\rho^2 - t^2 + x^2) + x + tf^{\frac{1}{2}}}{2\alpha^2(x^2 - t^2)}, \tag{A12}$$

with

$$f := \alpha^4(\rho^2 - t^2 + x^2)^2 + 2\alpha^2(\rho^2 + t^2 - x^2) + 1, \tag{A13}$$

and where for both solutions $y'_{1,2} = \beta$ and $z'_{1,2} = \gamma$ and where we have defined $\rho^2 = (y - \beta)^2 + (z - \gamma)^2$.

Comparing both solutions, we find that the solution that satisfies the second causality condition, $t' < t$, in the regions where the field does not vanish (I and II in Figure 1), is the one given by (A11).

We can now give the contravariant components of $\ell$:

$$\ell^t = \frac{\alpha^2 t(\rho^2 + t^2 - x^2) + t - xf^{\frac{1}{2}}}{2\alpha^2(t^2 - x^2)},$$

$$\ell^x = \frac{\alpha^2 x(\rho^2 + t^2 - x^2) - x + tf^{\frac{1}{2}}}{2\alpha^2(t^2 - x^2)}, \tag{A14}$$

$$\ell^y = y - \beta,$$

$$\ell^z = z - \gamma.$$

And also those of $u'$:

$$u'^t = \frac{\alpha^2 x(\rho^2 - t^2 + x^2) + x - tf^{\frac{1}{2}}}{2\alpha(x^2 - t^2)},$$

$$u'^x = \frac{\alpha^2 t(\rho^2 - t^2 + x^2) + t - xf^{\frac{1}{2}}}{2\alpha(x^2 - t^2)}. \tag{A15}$$

Finally, we check that:

$$\ell \cdot u' = -\frac{f^{\frac{1}{2}}}{2\alpha}. \tag{A16}$$

*Appendix A.3. Potential A in Cartesian Coordinates*

Once we have $\ell$ and $u'$ as a function of x, we obtain the following expressions for the covariant components of the potential using (8):

$$A_t = -\frac{q\left(\alpha^2 x(\rho^2 - t^2 + x^2) + x - tf^{\frac{1}{2}}\right)}{(x^2 - t^2)f^{\frac{1}{2}}},$$

$$A_x = \frac{q\left(\alpha^2 t(\rho^2 - t^2 + x^2) + t - xf^{\frac{1}{2}}\right)}{(x^2 - t^2)f^{\frac{1}{2}}}, \tag{A17}$$

with $\rho^2 = (y - \beta)^2 + (z - \gamma)^2$. This definition already induces a change to cylindrical coordinates, where $x$ is the height and $\rho$ the radial coordinate, so that the expressions do not depend on the azimuthal coordinate. The expressions are considerably simplified if, in addition, we use light-like coordinates defined as $u = t - x$ and $v = t + x$. In (21) we express the potential in this coordinate system.

*Appendix A.4. Graphic Representation of the Function f*

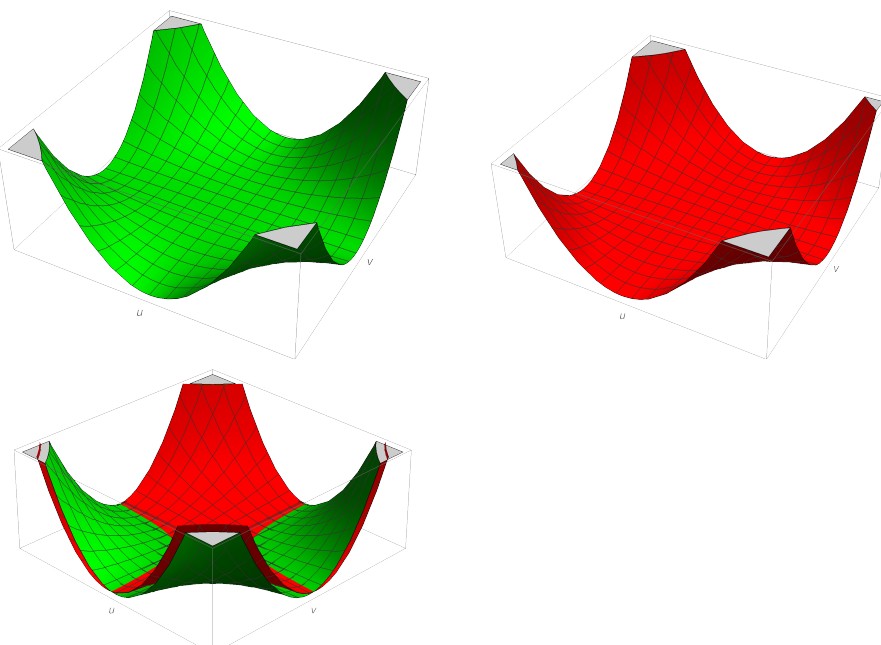

**Figure A1.** Representation of the 2-surfaces $f_\rho = f_\rho(u, v)$ considering $\rho$ as a parameter. In the figure on the top left (green) we have taken $\rho = 0$ and in the figure on the right, $\rho > 0$ (red). Both surfaces are shown below including the intersections between them. The third space-like coordinate $\chi$ has been omitted.

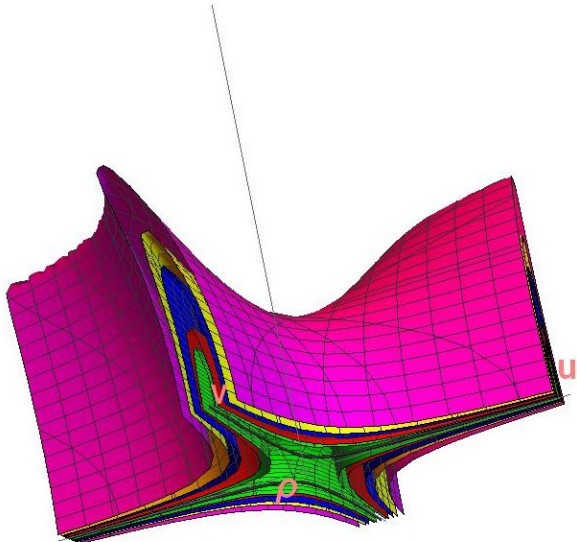

**Figure A2.** Perspective from below (below the $\rho = 0$ plane, with the $\rho$ axis being the vertical axis) of the level surfaces of $f$ for different values. The envelope surface (magenta colour) represents the highest value of $f$, the hyperbolas in black located in the foreground represent the lowest value $f = 0$, coinciding with the trajectory of the charge (here the symmetric trajectory at $v < 0$ is also represented).

## Notes

[1]  In fact, for Schott this solution was valid in $x + t \geq 0$.

[2]  These equations are reduced to the traditional ones when they refer to an inertial observer:

$$\nabla \cdot \vec{E} = \rho, \qquad \nabla \cdot \vec{H} = 0, \qquad \partial_t \vec{E} = \nabla \times \vec{H} - \vec{j}, \qquad \partial_t \vec{H} = -\nabla \times \vec{E}.$$

3    In the absence of *external* fields, see [10], pp. 614–615.

4    $D_r(\mathrm{x} - \mathrm{x}') = \frac{1}{2\pi}\theta(t - t')\delta[(\mathrm{x} - \mathrm{x}')^2]$.

5    See [9], p. 166.

6    This equation defines two hyperbolas, we take the one with $\mathrm{x}' > 0$.

7    With $u, v \in \mathbb{R}$, $\rho \geq 0$ and $\chi \in [0, 2\pi]$.

8    The coordinate representing the height $x$ being included in $u$ y $v$.

9    This interesting function is analysed in Section 4.6.

10    $\theta(v) = 1$ for $v \geq 0$ and $\theta(v) = 0$ for $v < 0$. Strictly speaking, this is the Heaviside *distribution*. See [11].

11    Where we have used the property $\partial_\mu \theta(v) = \delta_\mu^v \delta(v)$. See [11] for all the mathematical rigour that such a symbolic expression requires.

12    From (22) it can be proved that the limiting value of $f$ in the light-like 3-plane $v = 0$ is equal to $(1 + \alpha^2 \rho^2)^2$.

13    All the quantities that we are going to calculate from this field are valid in $v \neq 0$, in Section 5 we deduce the expressions of the electric and magnetic field for any value of $v$.

14    It being a simple 2-form.

15    Recall that, on the other hand, a singular field generally decomposes as $F = n \wedge k$, with $n$ light-like and $k$ space-like and $n \cdot k = 0$.

16    The factor $4\pi$ results from our choice of units.

17    Although the charge is at rest at $t = 0$, we must not forget that the field we measure at $t = 0$ originated at an earlier instant with the charge in motion.

18    Pauli relied on Born's solution (valid at $v > 0$).

19    pp. 225 et seq.

20    See [14], pp. 44–45, and also [15] for an analysis in three-dimensional notation.

21    $\partial_\rho \delta(v) = \partial_\rho \partial_v \theta(v) = \partial_v \partial_\rho \theta(v) = 0$. See [11] for all the mathematical rigour that such a symbolic expression requires.

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
