# Peer review of "Maxwell Field of a Charge in Hyperbolic Motion"

_universe, doi:10.3390/universe9060286_

Round 1
Reviewer 1 Report
The paper deals with the old problem of finding the electromagnetic radiation emitted by a uniformly accelerating charged particle in vacuum. The authors briefly introduced the well-known Liénard-Wiechert retarted potential solution and wrote the latter in a covariant form, and then they considered the case of accelerated motion of the source. The electromagnetic field and the stress-energy tensor in covariant form and using exterior calculus are all displayed, and a discussion of the validity region of the field solution is provided. The paper is well written and the calculations are shown in detail. Furthermore, as the authors pointed out, being in a covariant form the authors' results for flat spacetime might be generalized to curved spacetime.
The only thing I have against the present version of the paper is the awkward notation chosen by the authors. Below is the list of instances where such a notation is misleading, and where I suggest the authors bring some improvements to the manuscript before it becomes suitable for publication:
1) Sec. 1, line 22: The authors say "We denote our inertial observer by u=-dt". This statement simply doesn't make any sense to me.
2) Why don't the authors just use x and y to denote different position 4-vectors instead of using a prime like in y-x'(tau)? Using y-x(\tau) is just fine.
3) The authors decided to underline a letter whenever it refers to a Cartesian coordinate so that letters not underlined refer to position 4-vectors. Why then aren't the letters y' and z' underlined since they represent coordinates and not 4-vectors in Eqs. (14)-(18)?
4) Why is the null coordinate u underlined in Eq.(18), but v is not?
5) Why are x and u not underlined in Eq.(19) onward even though they still represent coordinates there?
6) The notation e for the electric field is not adequate as the components e^x and e^\rho look rather like exponential functions.
7) Why is a factor of 1/2 introduced in metric (19) even though the tilde on the tensor product symbol is said to mean symmetrized? The authors should specify in Sec. 1 which convention they use for symmetrisation.
8) The energy tensor T of Maxwell's field is rather called stress-energy tensor.
9) The words "Hyperbolic Charge" in the title are misleading as the hyperbolicity is not a property of the charge, but that of the motion of the charge.
If the authors do not like my suggestions for improvements, they should at least come up with a better and less confusing ones.
Reviewer 2 Report
The authors revisit the problem of electromagnetic field of a constantly accelerated charge using covariant differential forms. Their hope is that its covariant nature can be extended to this problem in curved space-time. In the differential form formalism components of the 4-potential are coefficients of the space-time differentials of dA where A is a Lorentz scalar, the Faraday tensor F is a differential 2-form, the source is a 4-current J, and Maxwell's equations are dF=0 and Div.F=-J. The retarded solution to Maxwell's wave equation is analyzed in detail. They separate F into coulombic and radiative parts where only the radiative part is acceleration dependent. To satisfy Maxwell's equations a source current Jv is added on the advanced hypersurface.
The presentation is interesting and mathematically rigorous and I recommend publication of this manuscript in its present form, but think that the Conclusions and perspective would benefit from additional discussion of the physics of their model.
